# Dynamic Mode Decomposition of Multiphoton and Stimulated Emission Depletion Microscopy Data for Analysis of Fluorescent Probes in Cellular Membranes

**DOI:** 10.3390/s24072096

**Published:** 2024-03-25

**Authors:** Daniel Wüstner, Jacob Marcus Egebjerg, Line Lauritsen

**Affiliations:** Department of Biochemistry and Molecular Biology, University of Southern Denmark, DK-5230 Odense M, Denmark; jegebjerg@bmb.sdu.dk (J.M.E.); linelauritsen@bmb.sdu.dk (L.L.)

**Keywords:** membrane, fluorescence, multiphoton microscopy, STED microscopy, computational microscopy, polarization, lipids, cholesterol

## Abstract

An analysis of the membrane organization and intracellular trafficking of lipids often relies on multiphoton (MP) and super-resolution microscopy of fluorescent lipid probes. A disadvantage of particularly intrinsically fluorescent lipid probes, such as the cholesterol and ergosterol analogue, dehydroergosterol (DHE), is their low MP absorption cross-section, resulting in a low signal-to-noise ratio (SNR) in live-cell imaging. Stimulated emission depletion (STED) microscopy of membrane probes like Nile Red enables one to resolve membrane features beyond the diffraction limit but exposes the sample to a lot of excitation light and suffers from a low SNR and photobleaching. Here, dynamic mode decomposition (DMD) and its variant, higher-order DMD (HoDMD), are applied to efficiently reconstruct and denoise the MP and STED microscopy data of lipid probes, allowing for an improved visualization of the membranes in cells. HoDMD also allows us to decompose and reconstruct two-photon polarimetry images of TopFluor-cholesterol in model and cellular membranes. Finally, DMD is shown to not only reconstruct and denoise 3D-STED image stacks of Nile Red-labeled cells but also to predict unseen image frames, thereby allowing for interpolation images along the optical axis. This important feature of DMD can be used to reduce the number of image acquisitions, thereby minimizing the light exposure of biological samples without compromising image quality. Thus, DMD as a computational tool enables gentler live-cell imaging of fluorescent probes in cellular membranes by MP and STED microscopy.

## 1. Introduction

Cellular membranes are highly complex assemblies of proteins and lipids which fulfill many functions as biological barriers in eukaryotic cells. Cell membranes not only compartmentalize metabolic processes but also act as signaling platforms harboring intricate molecular machineries for executing life at the subcellular level. These diverse functions are reflected in a very complex and diverse composition of not only proteins but also lipid species in cellular membranes. Phospholipids with differing head groups and acyl chains constitute the two leaflets of the plasma membrane (PM) and of subcellular membranes, defining unique territories to orchestrate membrane-associated biochemical processes [1]. The number of double bonds in the acyl chains of phospho- and sphingolipids and the cholesterol content of membranes also play important roles, as they dictate membrane fluidity, permeability, and bending flexibility (Figure 1A) [1]. Finally, the lipids in the bilayer can interact with membrane embedded and attached proteins. For example, the actin cytoskeleton underlying the inner leaflet of the PM (sketched in Figure 1A) can affect the diffusion of lipids, as shown by single-molecule tracking and stimulated emission depletion (STED) microscopy coupled to fluorescence correlation spectroscopy (FCS) [2,3,4]. To study lipid packing and the dynamics in such complex membrane assemblies, researchers often employ the fluorescence microscopy of suitable lipid probes. For example, dyes like Laurdan or Nile Red (Figure 1B), which embed faithfully into various cellular membranes, have a large change in dipole moment upon excitation, allowing for the sensitive detection of water penetration into membranes and thereby lipid packing via solvent-relaxation studies [5,6,7]. Probes such as Nile Red are also highly suitable for super-resolution microscopy like STED, as they are non-fluorescent in water but highly emissive in the lipid bilayer, such that they can be replenished from the aqueous phase into membranes upon light-induced photobleaching in the bilayer [8,9,10]. A challenge here is the tradeoff between resolution, which gets better for larger intensities of the STED laser, and the photon budget, which is reduced by photobleaching, thereby limiting the attainable signal-to-noise ratio (SNR). Fluorescent analogues of specific lipid species, such as cholesterol, can be used to follow the intracellular trafficking of these lipids in cells. Here, the chemical modification needed to obtain a fluorescence signal should be kept to a minimum to ensure sufficient resemblance of the natural lipid counterpart. One strategy is to use small organic dye molecules, such as 4,4-Difluoro-1,3,5,7,8-Pentamethyl-4-Bora-3a,4a-Diaza-*s*-Indacene (BODIPY) or nitrobenzoxadiazole (NBD), covalently linked to cholesterol, allowing for the visualization of the resulting fluorescent analogues conveniently via epifluorescence, confocal, or multiphoton (MP) microscopy [11]. The advantages of probes such as BODIPY-cholesterol (trade name TopFluor-cholesterol) for cellular studies are their high molecular brightness, relative photostability, and good two-photon absorption, allowing for prolonged time-lapse imaging and even super-resolution and single molecule microscopy (Figure 1B) [11,12,13,14]. Since already tiny structural changes in cholesterol change its properties significantly, a disadvantage of these tagged lipid probes is their limited resemblance of the natural sterols, making the interpretation of results challenging, and therefore requiring careful control experiments [11,15,16,17]. An alternative strategy is to use intrinsically fluorescent cholesterol analogues, which differ from cholesterol only minimally, since they do not contain attached fluorophores. Instead, analogues such as cholestatrienol (CTL) or dehydroergosterol (DHE, Figure 1B) contain few additional double bonds in the steroid ring system, giving them an intrinsic fluorescence while preserving the cholesterol-like biophysical and cell biological properties [11,15,18]. The disadvantages of these so-called polyene sterols are their low brightness and high bleaching propensity, as well as the need for ultraviolet (UV) excitation [11,18]. To image DHE in cells, both UV-sensitive wide-field and multiphoton (MP) microscopy have been employed [19,20]. We previously compared both imaging modalities and found that MP microscopy allows for 3D imaging of DHE in cells, while UV-sensitive wide-field imaging is faster and more suitable for routine investigations and for the discrimination of a probe from autofluorescence [21,22]. A particular challenge of MP imaging of DHE is the low SNR in each frame, requiring extensive image acquisition and image averaging to obtain single good-quality images [21]. On the other hand, using denoising algorithms such as PURE-LET denoising was found to be efficient in improving the quality of MP images of DHE [23,24]. Whether other routines are also suitable or even provide a better performance in regard to denoising the MP images of membrane sterols has not been investigated yet. 

MP microscopy does not only allow for live-cell imaging of sterol trafficking; it also allows us to assess the membrane probe orientation. By using two-photon polarimetry in which linearly polarized and pulsed infrared lasers are used to excite membrane probes with different orientations, the angle of membrane-embedded probes can be determined [25,26,27]. By using two-photon polarimetry, we were able to determine the orientation of BODIPY-moieties differently linked to cholesterol in model and cell membranes, and we could correlate probe orientation to the lateral diffusion dynamics of such analogues [13]. For an analysis of such data, Fourier-based signal decomposition is often employed [13,26,27,28]. This approach provides an in-depth analysis of the polarization data, even in a pixel-wise manner, but it does not account for unavoidable photobleaching during image acquisition. 

**Figure 1 sensors-24-02096-f001:**
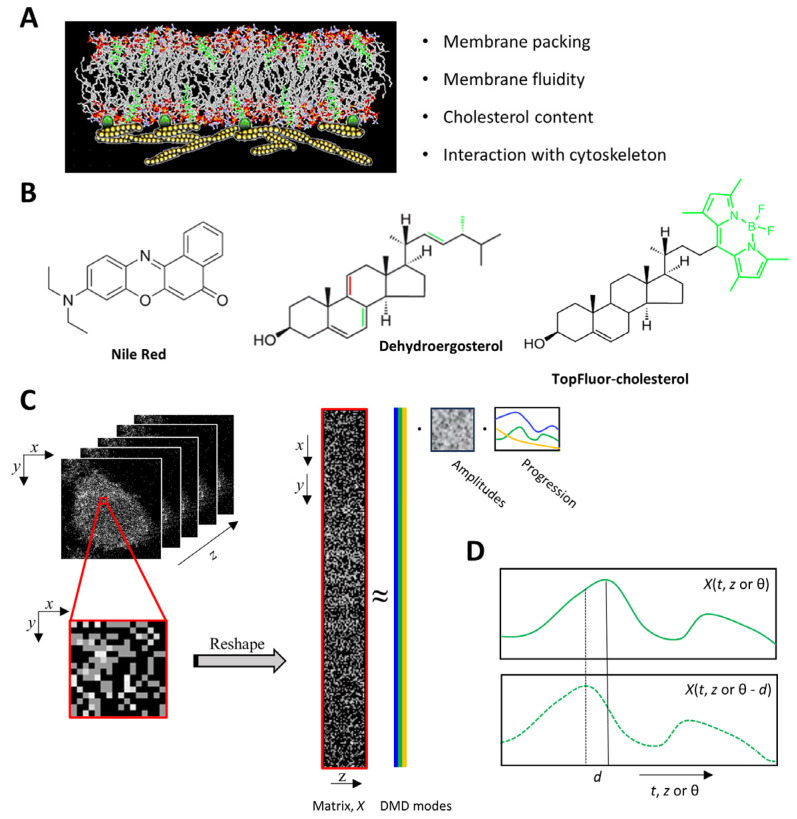
Principle of dynamic mode decomposition of fluorescence microscopy data for analysis of lipid probes in cell membranes. Cell membranes consist of a lipid bilayer made of phospholipids and sterols in which proteins are embedded (not shown) or adhered to the membrane, as sketched here for the cytoskeleton underlying the PM (**A**). The cartoon was adapted from an atomistic Monte Carlo simulation by the author [29]. Membrane properties, such as those listed on the right side, can be studied by fluorescence microscopy. Nile Red, dehydroergosterol (DHE), and TopFluor-cholesterol (TF-Chol) are membrane probes that are often employed to study cell membranes (**B**). How the sterol probes DHE and TF-Chol differ from cholesterol and ergosterol is shown in green and red, respectively. Multiphoton image acquisition of DHE as a function of z-coordinate is used for illustration of the DMD workflow (**C**). Reshaping of the image tensor is illustrated for a cropped part of the image, and the calculation of the DMD modes, amplitudes, and progression along the z-coordinate is shown. The principle of time-delay embedding is illustrated in (**D**). See main text for further details.

Dynamic mode decomposition (DMD) is a recently developed signal decomposition technique which allows for the dissection of complex dynamic processes, such as fluid flow [30,31]. The method has its origin in Koopman operator theory and is based on a singular-value decomposition (SVD) of the reshaped image data, providing a linear approximation of the often non-linear dynamics to study, for example, coherent flow patterns [32]. Based on these properties, DMD has also been used in computer vision and for the analysis of biomedical image data, for example, for background identification, for image segmentation or motion correction in Magnet resonance imaging (MRI), and for feature detection in positron emission tomography [33,34,35,36,37,38]. In live-cell imaging, matrix decomposition methods such as principal component analysis (PCA), non-negative matrix decomposition, or independent component analysis are often used to dissect and analyze dynamic processes [39,40]. In contrast, the use of DMD in microscopy is in its infancy and, to our knowledge, limited to our recent studies in which we showed the potential of this method for bleaching-based image segmentation and for the analysis of protein dynamics and aggregation in living cells upon fluorescence loss in photobleaching (FLIP) microscopy [41,42]. 

In this study, we demonstrate the power of DMD for high-fidelity reconstruction and denoising of MP and STED microscopy data of lipid probes in living cells. We show that DMD is on par with other efficient image-denoising methods, allowing for much improved SNR, even for challenging imaging applications, such as MP microscopy of DHE. Using a variant of DMD named higher-order dynamic mode decomposition (HoDMD) [43], we are also able to reconstruct and denoise 3D MP stacks of DHE and account for the small displacement of cells during image acquisitions. We show that HoDMD also enables one to reconstruct and analyze two-photon polarimetry data of TopFluor-cholesterol, thereby enabling a reliable assessment of probe orientation in membranes. Finally, we show that DMD of STED images of Nile Red not only provides much improved SNR but also allows us to predict unseen image frames along the optical axis. This enables one to reduce the light exposure of the samples and thereby to minimize sample damage. Together, our study demonstrates the large potential of DMD and its variants for the efficient postprocessing and analysis of live-cell fluorescence imaging data.

## 2. Materials and Methods

### 2.1. Cell Culture and Labeling

Immortalized normal human astrocytes (IM-NHAs) were purchased from Innoprot (cat. no. P10251) and were grown in astrocyte media (Innoprot, cat. no. P60101) in a humidified atmosphere supplemented with 5% CO_2_ at 37 °C. One day prior to imaging, the IM-NHAs were plated on glass slides in 35 mm microscope dishes (MatTek, cat. no. P35G-1.5-50-C). On the day of imaging, the cells were flushed once with M1 media (150 mM NaCl (Merck, Denmark, cat. no. 1.06404.1000), 5 mM KCl (Merck, cat. no. 104936), 1 mM CaCl_2_ (Merck, cat. no. 2382.1000), 1 mM MgCl_2_ (Merck, cat. no. 1.05833.1000), 5 mM glucose (Merck, cat. no. 1.08342.1000), and 20 mM HEPES (Sigma-Aldrich, Denmark, cat. no. H3375-100G), pH adjusted to 7.4) prior the addition of 2 µM Nile Red (Thermo Fischer Scientific, Denmark, cat. no. N1142) in M1 media, and imaging started immediately after, at room temperature.

### 2.2. Multiphoton Imaging of Fluorescent Cholesterol Analogues

All MP imaging data of fluorescent sterols, DHE, and TopFluor-cholesterol were generated and described in our previous publications, and the image data analyzed here are entirely from these studies [13,21]. For comparison with DMD methods, PURE-LET denoising of MP sequences of DHE was implemented using a plugin to ImageJ (https://bigwww.epfl.ch/algorithms/denoise/, accessed on 20 March 2024) [24]. For that, standard settings of three cycles and multi-frame analysis over three successive images were employed.

### 2.3. STED Microscopy of Nile Red

Confocal and STED microscopy was carried out with an Abberior Facility Line STED microscope (Abberior Instruments GmbH) with pulsed and circularly polarized lasers, using a UPlanSApo *x*100/1.40 NA oil objective and a pinhole size of 1 AU. Fluorescence of Nile Red was detected in the green channel between 498 and 551 nm, excited at 488 nm; and a red channel excited at 561 nm, with emission collected between 570 and 720 nm. Three-dimensional STED stacks were acquired using a 775 nm depletion laser with applied gating of 750 ps and a measurement width of 8 ns. The 3D stacks were acquired using a pixel size of 25 nm in *xy* and 50 nm in *z*.

### 2.4. Generation of Cell Phantoms for Benchmarking of DMD Performance

A cell phantom was generated with a Macro script in ImageJ based on the Macro ‘Sphere Builder’ developed by Drs. Hernan Sandra and Holder Lorens from Heidelberg University (wsr.imagej.net/macros/Sphere_Builder.txt, accessed on 20 March 2024). The cell phantom was convolved with the PSF using TensorFlow (https://www.tensorflow.org, accessed on 20 March 2024). Poisson noise-corrupted images were generated using the RandomJ plugin to ImageJ, generated by Dr. Meijering (University of New South Wales, Australia).

### 2.5. Analysis of MP and STED Microscopy Data by Dynamic Mode Decomposition

DMD is a matrix decomposition method requiring the image data to be gathered in series, i.e., as function of time, *t*, in time-lapse microscopy of *z*-position (i.e., position along the optical axis, *z*) in 3D imaging or as function of polarization angle, *θ*, in the case of two-photon polarimetry. The latter method is based on rotating a linearly polarized two-photon laser and recording fluorescence emission at each polarization angle [13,26]. Each of these methods generates a data cube with image coordinates, *x* and *y*, and the third axis representing the independent variables, i.e., *t*, *z*, or *θ*. Before using DMD, each of these datasets must be reshaped into a large matrix by concatenating the *x*- and *y*-coordinate into a column vector of *n* = *x* + *y* entries for each measurement. This gives one column at each given instance of time, z-position or polarization angle, i.e., for the independent variables, *t*, *z*, or *θ* (see Figure 1 for an illustration of this reshaping for the variable *z* and for *t*, as applied to bleaching-based image segmentation and FLIP microscopy, in [41,42]). Thus, for each instance of the independent variable in the original image sequence, one obtains a column in a large matrix of the form [32,36]:(1)$$X=\left[\bar{x_{1}},\;\bar{x_{2}},\;\cdots ,\bar{x_{k}},\ldots \bar{x_{m}}\right]$$

The index *k* = 1, …, *m* runs over all acquired images, i.e., snapshots in time, frames along the optical axis, or fluorescence intensities for each polarization angle. To model the progression of each of these image series, for example, for a time-series, one can, in principle, find a differential equation, but that is typically unknown for complex image data. Therefore, we want to approximate the progression from one image to the next directly from the data. In discrete time with steps, Δt, one can define the progression from state x(*k*·Δ*t*) = x_k_ to x_k+1_ as follows:(2)$$x_{k+1}=A\cdot x_{k}$$
where *A* is a matrix which describes the advancement of the system from image *x*_k_ to image *x*_k+1_. This matrix resembles the Koopman or transfer operator for measurements g(*x*_k_) = *x*_k_ [29]. For the analysis of 3D microscopy and two-photon polarimetry data, one replaces Δ*t* by Δ*z* and Δ*θ*, respectively. We want to approximate *A* solely from the given data. For that, we define the discrete time-shifted states of our system as two new matrices, X_1_ and X_2_ ∈ *R*^n·(m−1)^:(3)$$X_{1}=\left[\bar{x_{1}},\;\bar{x_{2}},\;\cdots ,\bar{x_{m-1}}\right]$$
and
(4)$$X_{2}=\left[\bar{x_{2}},\;\bar{x_{3}},\;\cdots ,\bar{x_{m}}\right]$$

With that, the system corresponding to Equation (2) becomes *X*_2_ = *A*·*X*_1_, from which we want to find the matrix, *A*. In the imaging applications considered here, the data matrices, *X*_1_ and *X*_2_, have many more rows *n* (i.e., pixels for each image) than columns (*m* − 1) (i.e., time points, *t*; z-stack positions, *z*; or polarization angles, *θ*; Figure 1C). Thus, we cannot invert *X_1_* directly but find *A* by minimizing the Frobenius norm, ${\left\|\cdot \right\|}_{F}$, instead [30]:(5)$$A∶=argmin{\left\|X_{2}-A\cdot X_{1}\right\|}_{F}=X_{2}\cdot X_{1}^{inv}$$
where $X_{1}^{inv}$ represents the pseudoinverse of the first snapshot matrix, which we find by using an SVD of *X*_1_ into unitary matrices U ∈ *R*^n·(m−1)^ and V* ∈ *R*^n·n^, with singular values in the diagonal matrix Σ ∈ *R*^n·(m−1)^:*X*_1_ = *U*·*Σ*·*V** (6)

There are, at most, (*m* − 1) non-singular values and corresponding singular vectors, and therefore, the matrix *A* will have, at most, rank (*m* − 1). Thus, we have maximally *m* snapshots, which represent the measurements, i.e., the image acquisitions at a given time, *t*; axial position, *z*; or polarization angle, *θ*. Accordingly, maximally *m* dynamic modes can be determined by classical DMD, which is sometimes called the spectral complexity of the system [43]. In practice, one often calculates the modes only up to rank *r* < (*m* − 1): one approximates the system matrix, *A*, by projecting it onto the left singular vectors, i.e., the column vectors of U. This gives a much smaller matrix, A’, of maximal size *r* times *r* via a similarity transformation [30,34]:*A*’ = *U*’*·*A*·*U*’ = *U*’*·*X*_2_·*V*’·*Σ*’^−1^
(7)
where *U*’, *V*’, and *Σ*’ are rank *r* approximations of the full matrices, *U*, *V*, and *Σ*. The complex conjugate transposed matrix is indicated by *, which is equal to the transpose for a real matrix, as is the case for our imaging data:*A*’ = *U*’^T^·*A*·*U*’ = *U*’^T^·*X*_2_·*V*’·*Σ*’^−1^
(8)

Since *A* and *A*’ are similar, they have the same eigenvalues, and we can analyze the progression of the full system from one snapshot to the next by analyzing the reduced matrix, *A*’. Thus, the similarity transformation of Equation (7) corresponds to a dimension reduction, reducing the size of the system matrix from *A* ∈ *R*^(m−1)·(m−1)^ to *A’*∈ *R*^r·r^ [30]. This step is essential in achieving the denoising performance of DMD, as further shown below.

To obtain the spectral decomposition of the reduced system matrix, *A*’, one finds the eigenvalues, *λ_j_*, and eigenfunctions, *φ_j_*, for each DMD mode *j* by solving the corresponding eigenvalue problem. This leads to a discrete system: (9)$$x_{k}=\sum_{j=1}^{r}\varphi_{j}\cdot \lambda_{j}^{k-1}\cdot b_{j}$$

For a continuous system, e.g., in time, one can rescale the eigenvalues according to *ω* = ln(*λ*/Δ*t*), such that Equation (9) can be written as follows [33]:(10)$$x(t)=\sum_{j=1}^{r}\varphi_{j}\cdot e^{\omega_{j}\cdot t}\cdot b_{j}$$
where *x*(*t*) is a vector of images (x, y index omitted for brevity) as a function of time, *t*, but for our purpose, the independent variable *t* could be replaced by *z* for DMD along the optical axis (i.e., *x*(z)) or by *θ* for decomposing two-photon polarimetry data (i.e., *x*(*θ*); see Figure 1C). Thus, Equation (10) describes here the temporal evolution of each dynamic mode, *φ*_j_, which is a function of space, only. The mode amplitudes, *b*_j_, are spatial weighting matrices, i.e., functions of pixel coordinates (x, y), accounting for the initial intensities. DMD is related to the discrete temporal Fourier transformation (DFT), but in contrast to DFT, which only decomposes a signal into oscillating modes, DMD also accounts for growing or decaying signals [44]. In fact, when subtracting the mean before carrying out a DMD and thereby accounting for continuous intensities’ decreases or increases, DMD becomes identical to the DFT [44]. The eigenvalues of the DMD in Equation (10), *ω*_j_, can therefore be considered the complex Fourier modes of the system, where the real part describes the mode’s decay or increase, while the imaginary part describes mode oscillations. An important condition for employing DMD to spatiotemporal experimental data is that there are enough snapshots (i.e., measurements) to capture the full spectral complexity of the data. This is often not the case, particularly when the data contain transients or other non-linear dynamic contributions, which would require many dynamic modes to be described accurately. In these cases, the number of linearly independent DMD modes attainable from the data is not sufficient to capture their full complexity, resulting in a poor reconstruction quality. This has been shown for even simple systems, such as a standing wave, but also for highly non-linear and chaotic systems [45]. To overcome this limitation of standard DMD, the idea of time-lagged embedding can be employed, in which time-lagged versions of the snapshots are used to increase the dimension of linearly independent basis functions [43,45]. 

Specifically, a particularly attractive variant of DMD, named HoDMD, can be used, which is based on a higher-order Koopman assumption [43]:(11)$$x_{k+d}=A_{1}\cdot x_{k}+A_{2}\cdot x_{k+1}+..+A_{d}\cdot x_{k+d-1}$$
where *k* runs from 1 to *m*-*d*, where *m* again is the number of available snapshots, and *d* is the set time delay. After applying the SVD of Equation (6) and the rank-*r* approximation of Equations (7) and (8), one uses the representation of Equation (11) to describe the evolution of the system as follows:(12)$${\tilde{X}}_{k+1}=A\prime \cdot {\tilde{X}}_{k}$$

The reduced matrices for the snapshots and the evolution of the system read as follows [43]:(13)$${\tilde{X}}_{k}=\left(\begin{array}{c} x_{k}\\ x_{k+1}\\ \ldots \\ x_{k+d-1} \end{array}\right),\;\;A\prime =\left(\begin{array}{cccc} I & 0 & \ldots & 0\\ 0 & I & \ldots & 0\\ \ldots & \ldots & \ldots & 0\\ A_{1} & A_{2} & \ldots & A_{d} \end{array}\right)$$

Thus, the key idea of HoDMD is to use time-lagged snapshots, which allow us to obtain sufficient information for reconstructing systems with high spectral complexity from a limited number of snapshots. 

## 3. Results

### 3.1. High-Fidelity Reconstruction and Denoising of Multiphoton Microscopy Data by DMD

MP imaging of fluorescent cholesterol analogues, such as DHE, is plagued by a very low SNR due to the low cross-section for simultaneous absorption of several photons [20,21,46], as exemplified in Figure 2. This low excitation probability of DHE from the simultaneous absorption of several photons results in shot noise-limited images, as the emission of photons follows a Poisson process. As a consequence, the image noise cannot be considered as additive Gaussian. Applying DMD to an image series of DHE-stained CHO cells results in a very efficient denoising for each frame. Only one dynamic mode with a real eigenvalue describing the slight intensity decay due to photobleaching is sufficient to reconstruct the data with high accuracy (Figure 2A,B). A comparison with a widely used denoising method, PURE-LET denoising, shows that DMD provides a higher peak signal-to-noise ratio (PSNR) over the entire image series (Figure 2C). The highly efficient denoising is a consequence of the simple singular-value spectrum of these MP image series. Indeed, the snapshot matrix of this image series is dominated by one singular value capturing the majority of the variation in the data (Appendix A). Both DMD and HoDMD with automatic rank truncation are equally efficient in detecting the low-rank structure in the image data, allowing for the separation of image content from noise. This is illustrated in Appendix A, where one sees that the image noise corresponds to a plateau in singular values, which can be efficiently separated from the image features by the SVD.

Image reconstruction and denoising by DMD is even possible in the case of slight displacement of subcellular structures during MP imaging. This is shown for a DHE-labeled hepatocyte-like HepG2 cell in Figure 3. In contrast to still objects, flow-like movement of subcellular structures is captured in the imaginary part of the eigenvalues and dynamic modes.

This is shown in Figure 3B–E, where moving DHE-containing vesicles result in real and imaginary mode contributions. Again, the real part of the corresponding eigenvalues describes photobleaching-induced intensity decays, while the imaginary part is due to particle displacements. These results clearly show that DMD can capture the dynamics of MP image time series which have simple dynamic behavior, such as slight acquisition-induced photobleaching or small object displacements. This supports our previous analysis [41], here, for a challenging application with very low SNR. To ensure a good performance of DMD in this case, we used the noise-robust optimal DMD method, in which the error between the modes and all the snapshots is minimized, a procedure which provides improved estimates of amplitudes in the presence of noise [47,48]. Together, the results of Figure 2 and Figure 3 demonstrate the efficiency of DMD in denoising Poisson-corrupted images with very low SNR, which makes the method a suitable post-processing step for live-cell imaging based on photon-counting detectors.

A particular advantage of MP imaging is the ability to visualize subcellular structures in all three dimensions. This is due to the intrinsic sectioning capability of the method combined with the use of infrared light, allowing for deep specimen penetration [46]. To assess the potential of DMD for reconstructing the MP image data of DHE-labeled cells in 3D, we analyzed HepG2 cell couplets forming a central intercellular membrane compartment called a biliary canaliculus (BC; Figure 4). The formation of a BC indicates polarization of the cells, and this compartment resembles the canaliculi biliferi of the liver, into which the bile fluid is secreted [49]. We showed previously that fluorescent sterols such as DHE are efficiently transported to the BC in HepG2 cell couplets, resembling the pathway for cholesterol secretion into the bile [50]. The DHE-labeled BC is clearly visible between two cells in 3D image series acquired by MP microscopy (Figure 4A). When applying DMD to these 3D stacks, efficient noise removal is observed; however, the reconstruction quality is limited, because the algorithm wrongly assigns the BC structure to almost all frames (Figure 4A upper and middle rows). This is a sign of mode mixing, i.e., the inability of DMD to correctly dissect the 3D image signal into separate spatial modes and their associates’ evolution along the optical axis. The limited reconstruction quality of standard DMD is also clearly seen in the integrated intensity plotted as a function of the frame number along the 3D stack in Figure 4B. In contrast, using HoDMD with delay embedding of *d* = 10 results in a much better reconstruction quality; HoDMD correctly assigns the BC structure to the central frames only and gives an integrated intensity closely coinciding with the experimental data (Figure 4A, lower rows; and Figure 4B). The eigenvalue spectrum reveals that HoDMD but not DMD contains complex eigenvalues with a relatively large imaginary contribution (compare Figure 5C,D). These complex eigenvalues determined by HoDMD can capture the non-monotonic progression of the DHE intensity along the optical axis adequately, thereby preventing mode mixing (Figure 4B). The integrated intensity additionally drops along the optical axis, which is likely due to the photobleaching of the sterol probe during acquisition. The overall trend of decaying intensity along the optical axis is correctly captured by both the standard DMD and HoDMD, but only HoDMD can account for the precise intensity profile and the specific image features in the 3D image stack. 

This important difference in the performance of both methods resembles that found previously when describing standing waves, for which standard DMD fails, while HoDMD gives correct results [43]. The increased data matrix of HoDMD augmented with z-shifted versions of the 3D image data provides more linearly independent rows to provide pairs of complex conjugated eigenvalues needed to describe the oscillation in the data, thereby preventing mode mixing from occurring [30,43]. The ability of the two DMD variants to dissect 3D intensity profiles was further assessed via an analysis of a theoretical point spread function (PSF) model of the microscope. For the simplicity of the analysis, we chose the PSF for an epifluorescence microscope system [51]. The reconstruction of this PSF model by HoDMD coincides well with the original data, as shown in 2D and 1D intensity profiles in Appendix A. Some deviation is found in out-of-focus intensity profiles, which, however, have drastically reduced intensity. For proper reconstruction of the data, more than 20 dynamic modes are included when choosing the automatic truncation method based on a hard threshold of the singular-value spectrum [52]. The HoDMD reconstruction provides a correct integrated intensity profile at the in-focus position, while some oscillatory behavior is found outside of the focal plane. In contrast, standard DMD completely fails to reconstruct the PSF profile, again showing that data augmentation via the embedding of z-shifted snapshots results in a much-improved DMD performance. 

To assess the effects of resolution and image noise on DMD performance, we carried out additional experiments with synthetic image stacks of a 3D cell phantom (Appendix A). Prior to convolution, to mimic microscope conditions, this phantom contains very sharp intensity transitions. While this is an extreme scenario, one that is not observed in experimental data, it allows us to find the limits of DMD methods in reconstructing 3D image stacks. HoDMD can reconstruct the original data of cell phantoms with only minor artifacts along the optical axis, particularly when using the full-rank snapshot matrix. In contrast, DMD is unable to reconstruct these data. After convolving the phantom image stack with a theoretical PSF, both DMD and HoDMD with automated rank truncation are able to reconstruct the input data very well (Appendix A). This shows that the limited resolution obtained in microscopy experiments has a smoothening effect on intensity transitions, thus facilitating the analysis by DMD. A convolved cell phantom corrupted with Poisson noise can be reconstructed and denoised by HoDMD, while standard DMD fails to provide adequate reconstruction quality, particularly along the optical axis (Appendix A). We observed that these different abilities of DMD versus HoDMD to reconstruct the image data depend on the singular-value spectrum of the image stacks. Convolving a cell phantom with a theoretical PSF caused a more rapid decay of singular values compared to non-convolved cell phantoms. It is easier to find a low-rank representation for a matrix with rapidly decaying singular-value spectrum, and this explains why both DMD variants can better capture the image features in the convolved cell phantom data. Supporting that notion, the rank of the image matrix for frame fifty-seven, which exemplifies the reconstruction quality, as shown in Appendix A, is seventy-nine in the non-convolved phantom image stack but forty-five in the convolved input data. In contrast, for the first image frame, which does not contain any structure, the rank is zero for the non-convolved but nineteen for the convolved image. Thus, lowering the image resolution by blurring redistributes the image information to neighboring planes. This makes it easier to obtain a reliable low-rank approximation of the Koopman operator matrix, explaining the better performance of the DMD algorithms on the blurred image data. In convolved 3D images corrupted with Poisson noise, the singular-value spectrum shows a plateau, as we observed in the experimental data (e.g., Appendix A). HoDMD outperforms standard DMD in capturing the image features in the noisy 3D cell phantoms adequately. 

### 3.2. Reconstruction of Two-Photon Polarimetry Data of Membrane Probes by HoDMD

The orientation of fluorescent probes in biological membranes can be determined using polarized two-photon excitation. In this method, named two-photon polarimetry, the fluorescence response of a probe to excitation by a linearly polarized two-photon laser as a function of the rotation angle, θ, is recorded [26]. The two-photon polarimetry measurements of TopFluor-cholesterol in BHK cells after disruption of subcortical actin enable one to determine the probe orientation in the PM from the sine-shaped fluorescence response (Figure 5A,B). Both DMD and HoDMD result in efficient denoising of the image data and account for the overall oscillation in the signal intensity (Figure 5A). There are two intensity peaks around 100 and 275°, suggesting that the probe is oriented predominantly perpendicular to the membrane plane (Figure 5B), as we confirmed using a discrete Fourier decomposition of the signal [13]. In addition, there is a slight intensity decay due to photobleaching during recording of the image stacks that cannot be accounted for by a standard Fourier analysis. In contrast, both DMD and HoDMD identify this intensity decay, as reflected by eigenvalues slightly smaller than unity giving exponentially decaying components of the associated ‘dynamics’ (Figure 5C,D). While DMD can capture the overall orientation-dependent probe fluorescence adequately, it overestimates the intensity decay, resulting in a too large drop in amplitude for the second peak around 260°, as inferred from the integrated intensity as a function of the polarization angle (Figure 5B, green line). There are two oscillatory and two exponentially decaying dynamic modes determined by DMD, as can be seen from the eigenvalue spectrum (Figure 5C) and the plots of the modes as a function of the polarization angle (Appendix A). In contrast, HoDMD accurately captures both intensity peaks when either six or nine shifts are used (*d* = 6, 9; Figure 5B, cyan and violet line). Due to the delay embedding, HoDMD provides seven eigenfunctions to dissect and reconstruct the data, while standard DMD only provides four modes (Figure 5C,D and Appendix A). For higher values, e.g., *d* = 20, HoDMD can capture even slight intensity variations along the stack, but since the data are noisy, this likely resembles overfitting (Figure 5B, yellow line).

In summary, delay-embedding of microscopy image stacks allows for a spatiotemporal dissection and high-fidelity reconstruction of the image data using HoDMD. The reconstructed microscopy data contain less noise than the original recordings thanks to the rank truncation of the approximated Koopman operator [43]. That is, only relevant modes describing the evolution of the image data along the z-axis are retained, while contributions to the signal with small singular values are discarded. The latter effect is very useful for removing noise-induced artefacts in the microscopy data, as illustrated by the MP microscopy data shown above.

### 3.3. Reconstruction of 3D-STED Microscopy Image Stacks by DMD

STED microscopy of Nile Red provides information about subcellular membranes with increased resolution compared to confocal microscopy [9,10]. This is achieved by the use of a red-shifted donut-shaped STED laser, which efficiently shuts off the fluorescence of molecules in the periphery of the area excited by the illumination laser. The focal volume can be narrowed down by increasing the power of the STED laser, and in our implementation, this is achievable in all three dimensions (3D-STED). 

When labeling human astrocytes with Nile Red, individual vesicles; lipid droplets; and mitochondria, including mitochondrial ultrastructure, can be discerned in 3D-STED image stacks of subcellular regions (Figure 6). Both DMD and HoDMD can almost perfectly reconstruct these data, as seen by the visual inspection of the reconstructed stacks, as well as from the quantification of the mean squared error (MSE) and structural similarity index measure (SSIM) between selected image frames (Figure 6A,B). The reason for the comparably good performance of both DMD methods lies in the rapidly decaying singular-value spectrum for these data (see Appendix A). This allows for finding a good low-rank approximation of the matrix *A* in Equation (2) and for extracting the dominant modes, because the majority of the relevant image information is contained in few singular values. For both DMD and HoDMD, STED image stacks have reduced noise levels, and the shape of organelles is well preserved after reconstruction, also along the optical axis (Figure 6B,C). The integrated intensity of the reconstructed image stacks calculated along the optical axis is very well aligned with that of the raw data, showing that both DMD and HoDMD have a good performance in regard to reconstructing STED image data (Figure 6D). Also, both methods find similar eigenvalues and dynamic modes, with HoDMD being slightly more accurate due to its ability to identify additional eigenfunctions of the approximated Koopman operator compared to DMD (Figure 6B–D and Appendix A). 

### 3.4. Interpolation of Missing Frames in 3D-STED Microscopy Image Stacks by DMD

The dynamic modes determined by DMD methods represent basis functions in a vector space whose linear combinations allow us to describe the progression of image data from one frame to the next. Therefore, this representation should also allow for the prediction of the image frames not available in the original data. To test this idea, we reconstructed the same 3D-STED image stack as described above, but with only every third frame in the input data. Thus, the new snapshot matrix, X’_1_, contains only every third column compared to the original snapshot matrix, X_1_, which consisted of 40 columns (i.e., reshaped image frames; see Equation (3)). Applying DMD, we aim for not only reconstructing the down-sampled STED image stack but also for interpolating the missing frames using the predictions based on the inferred transition matrix, *A*’ (see Equations (8)–(10)). As shown in Figure 7A, DMD can interpolate the missing frames with high quality. The reconstruction is of only slightly lower accuracy compared to that achievable by linear interpolation of the missing frames, as inferred from the MSE and SSIM of selected views of the 3D stacks in Figure 7B,C. The integrated intensity of the DMD reconstructed volume resembles the original STED image stack containing all frames closely. The linear interpolation of missing frames is again slightly more accurate (Figure 7D). To further compare both methods, we calculated the MSE and SSIM just for the interpolated frames along the entire stack and found that linear interpolation is more accurate in the first half of the STED images (Figure 7E). In the second half of the reconstructed volume, the MSE values for both interpolation methods approach each other, while DMD scores higher for the SSIM. The difference between both quality measures could be due MSE being more sensitive to noise levels. Since DMD is denoising the data, it can have higher MSE values compared to linear interpolation. 

SSIM is more sensitive to image distortions, and here, both methods show a comparable performance, particularly in the second half of the reconstructed volume. The axial resolution in our 3D-STED system is about 110–120 nm in the red channel. The original images were sampled every 50 nm, while the down-sampled stack contained images only every 150 nm. Thus, both interpolation methods can restore Nyquist sampling without compromising image quality significantly.

## 4. Discussion

Live-cell fluorescence imaging of membrane dynamics is an important tool to understand lipid and protein trafficking and to discern signaling processes in living cells. Cellular membranes are highly heterogeneous and can flexibly adapt to the complex requirements of their environment. A fundamental challenge in the imaging of membranes and, in particular, of membrane lipids is the need to label the studied molecules without altering their behavior. This demand often poses limitations on the fluorescent moieties requiring sensitive monitoring with an optimized photon budget. At the same time, there is the wish to increase the optical resolution, thereby obtaining insight into membrane dynamics at the nanoscale. Here, techniques such as STED microscopy are very powerful, but they pose additional demands on the used fluorescent probes, namely efficient photoswitching for extended times without photodegradation. Decreasing the incident light dose used for probe excitation is a valid strategy to alleviate these problems, but that comes at the price of a reduced signal-to-noise ratio for the obtained image stacks. Denoising the image data is therefore an essential step in the analysis of the live-cell microscopy of membrane dynamics. Various approaches have been developed for image denoising in microscopy, including total variation and sparse reconstruction, Fourier-space and wavelet-based denoising (e.g., in [53,54,55,56]), and dictionary-based approaches [57]. Deep learning methods for denoising are based on convolutional neural networks (CNNs), even in the absence of ground-truth data [58], variational autoencoders [59], and diffusion-based probabilistic generative models [60]. Here, we show that DMD—particularly HoDMD—is a suitable alternative to such methods in the case that the image data consist of stacks in which frames were acquired over time, as in time-lapse microscopy; along the optical axis, as in 3D imaging; or for different orientations of the excitation light, as in fluorescence polarimetry. We show that the coherent structures acquired by these microscopy modalities can be adequately captured by the DMD variants. The SVD with rank truncation results in the retainment of relevant modes, while efficiently removing image noise, even for very challenging conditions, such as MP microscopy of weakly fluorescent membrane probes, like DHE. Using synthetic cell phantoms, we show that the singular-value spectrum of the image data is an important factor for the performance of the DMD variants; lowering the image resolution relative to the spatiotemporal sampling rate will cause a more rapid decay of the singular values, which facilitates a DMD analysis of the microscopy data. This observation is in line with previous studies, showing that DMD fails if the data contain abrupt changes or transients, which we have in the synthetic non-convolved data [45]. On the other hand, we show that, for down-sampled 3D image stacks containing smooth image features, DMD is able to not only reconstruct and denoise the image data but also to interpolate missing frames. This important property will allow one to reduce the exposure time of the sample by acquiring fewer frames along the optical axis, since a full 3D volume can be generated by up-sampling the image data via DMD reconstruction. In this regard, however, DMD performance is not better than linear image interpolation, and both methods can be used to reduce the sampling frequency of 3D image volumes, thereby allowing for gentler 3D microscopy with less light-induced damage for the sample. Further improvements of DMD could be directed towards constraining the search for eigenfunctions of the Koopman operator by including information about the physics of the underlying processes. This approach has been successfully employed as a variant of DMD for the analysis of mechanical systems and fluid dynamics [61]. Physics-informed DMD could employ a physically meaningful description of the matrix *A* for each of the aforementioned scenarios, e.g., for the advancement of the image structures in fluorescence polarimetry or 3D imaging. Informing computational models with the underlying physics is an emerging approach for improving image recovery in various microscopy modalities [62], and we believe that the same strategy applied to DMD can contribute to its widespread use in the bioimaging community in the future.

## 5. Conclusions

Cell membranes are heterogeneous and highly dynamic protein–lipid assemblies, whose investigation via quantitative microscopy techniques profits from advanced computational tools. In this article, we show that DMD and its variant HoDMD can be used to decompose, denoise, reconstruct, and interpolate the multidimensional fluorescence imaging data of membrane probes. This allows for high-fidelity non-linear and super-resolution microscopy with reduced light exposure, thereby preventing artefacts introduced by cell stress due to extensive illumination.

## Figures and Tables

**Figure 2 sensors-24-02096-f002:**
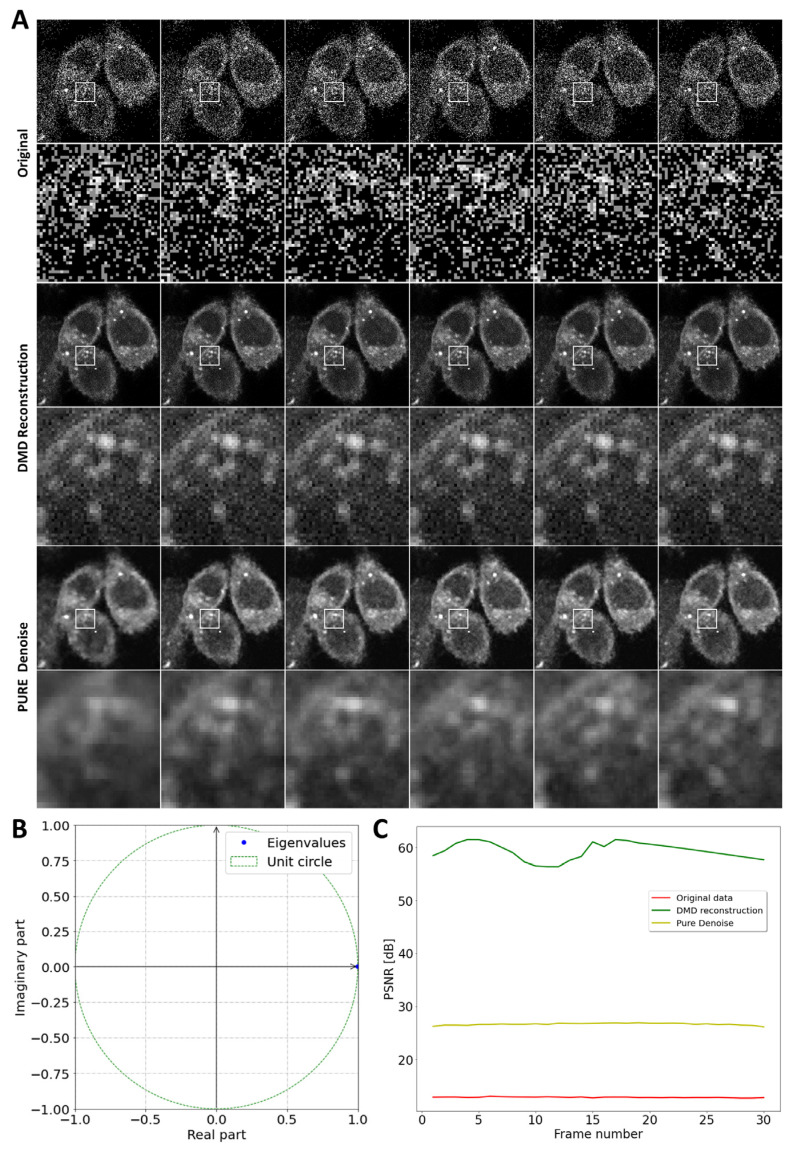
Reconstruction and denoising of MP image series of DHE-labeled cells. Consecutive raw images (upper panels) reconstructed using DMD (middle panels) or denoised using PURE-LET denoising (lower panels) are shown (**A**). The inset box is shown as a zoomed-in version underneath the respective panel. Eigenvalues recovered by DMD are plotted on the unit circle (**B**). The PSNR is plotted as a function of frame number for the original image series (red line), for the DMD reconstruction (green line), and for the PURE denoised stack (yellow line) (**C**). Original data shown in (**A**) were generated as described in [21].

**Figure 3 sensors-24-02096-f003:**
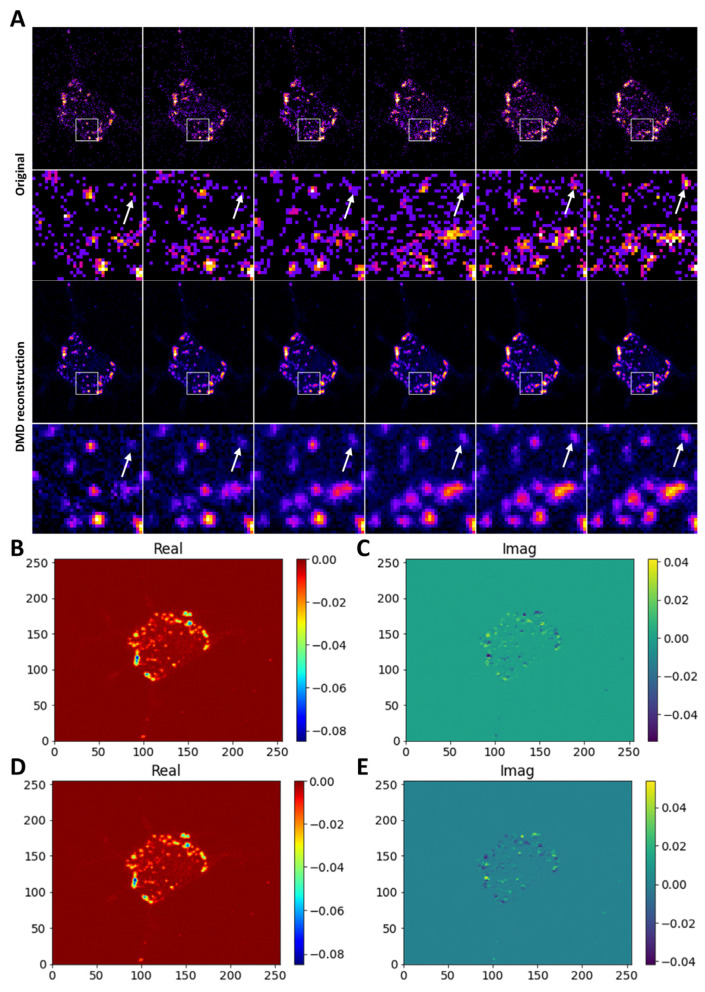
DMD reconstruction of MP images in the presence of vesicle displacement. CHO cells stained with DHE were imaged on an MP microscope and analyzed using DMD. Raw images are of low photon counts, as shown in the upper row, while the DMD reconstruction is shown in the lower row (**A**). The inset box is shown as a zoomed-in version underneath the respective panel, with arrows pointing to a moving vesicle. Real and imaginary part of the first mode (**B**,**C**) and the second mode (**D**,**E**) as obtained by DMD are shown. Non-zero values in the imaginary part indicate vesicle displacements, as accurately captured by DMD. Original data shown in (**A**) were generated as described in [21].

**Figure 4 sensors-24-02096-f004:**
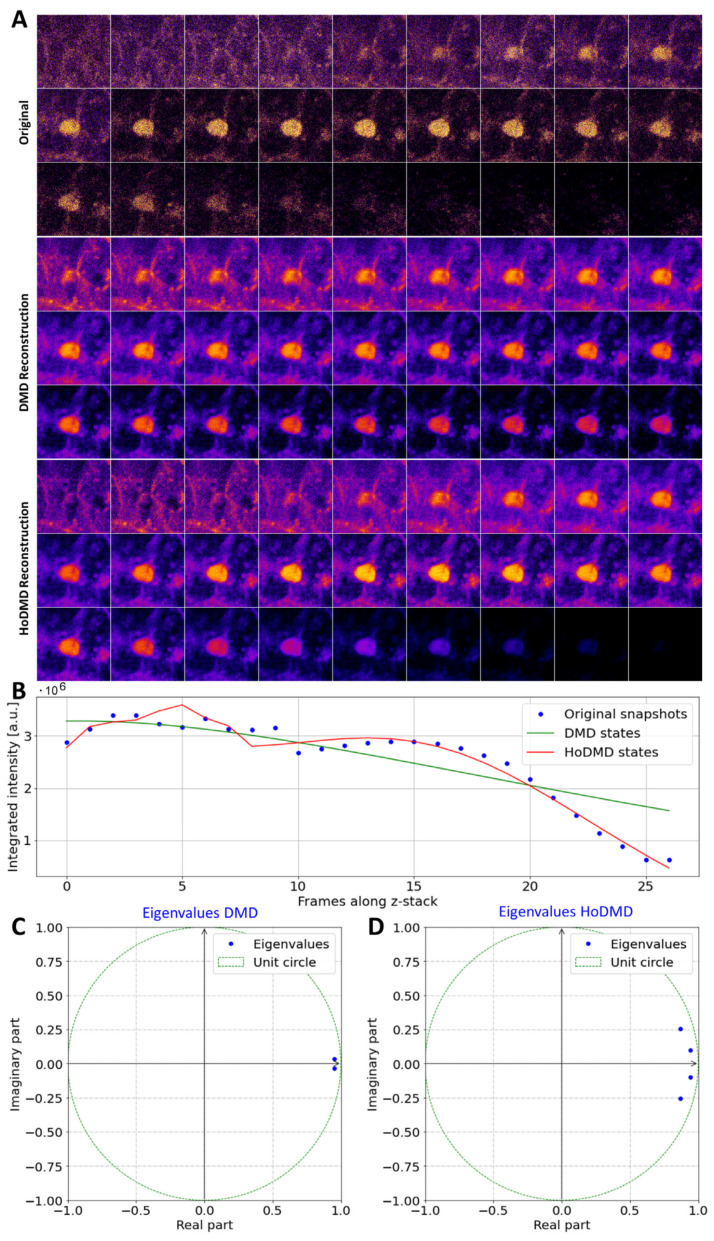
Comparison of DMD and HoDMD in reconstructing 3D MP image stacks of DHE-labeled cells. HepG2 cells forming an apical biliary canaliculus containing DHE were imaged by MP microscopy. Montages of unprocessed MP (upper row, ‘Original’), DMD reconstruction (middle row), and HoDMD reconstruction (lower row) in (**A**). The integrated intensity is shown for the original snapshots (blue symbols), the DMD reconstruction (green line), and the HoDMD reconstruction (red line) in (**B**). Eigenvalues recovered by DMD (**C**) or HoDMD (**D**) are plotted on the unit circle. Original data shown in (**A**) were generated as described in [21].

**Figure 5 sensors-24-02096-f005:**
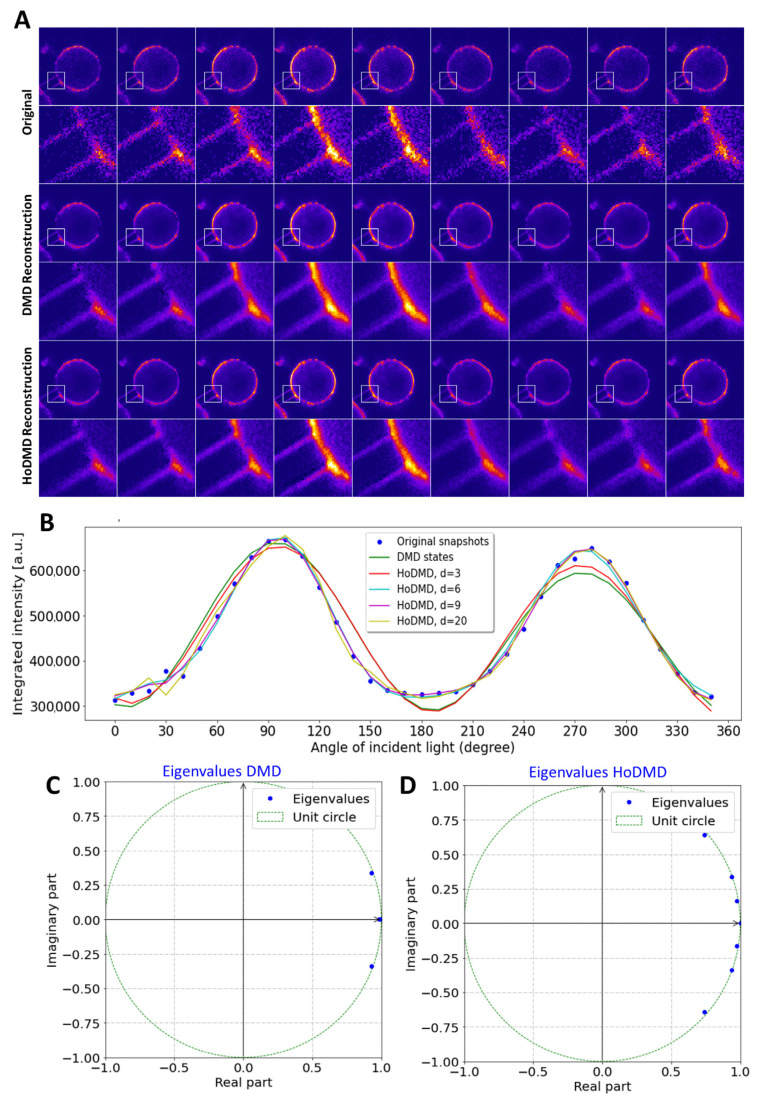
Comparison of DMD and HoDMD in reconstructing MP polarimetry image stacks of cells labeled with TF-Chol. CHO cells were stained with TF-Chol after disrupting actin with cytochalasin D, as described in the Materials and Methods and in [13]. Montages of unprocessed MP (upper row, ‘Original’), DMD reconstruction (middle row), and HoDMD reconstruction (lower row) are shown with zoomed box underneath each row in (**A**). The integrated intensity is shown for the original snapshots (blue symbols), the DMD reconstruction (green line), and the HoDMD reconstructions with increasing number of delays, *d* (*d =* 3, red line to *d =* 20, yellow line) in (**B**). Eigenvalues recovered by DMD (**C**) or HoDMD (**D**) are plotted on the unit circle. Original data shown in (**A**) were generated as described in [13].

**Figure 6 sensors-24-02096-f006:**
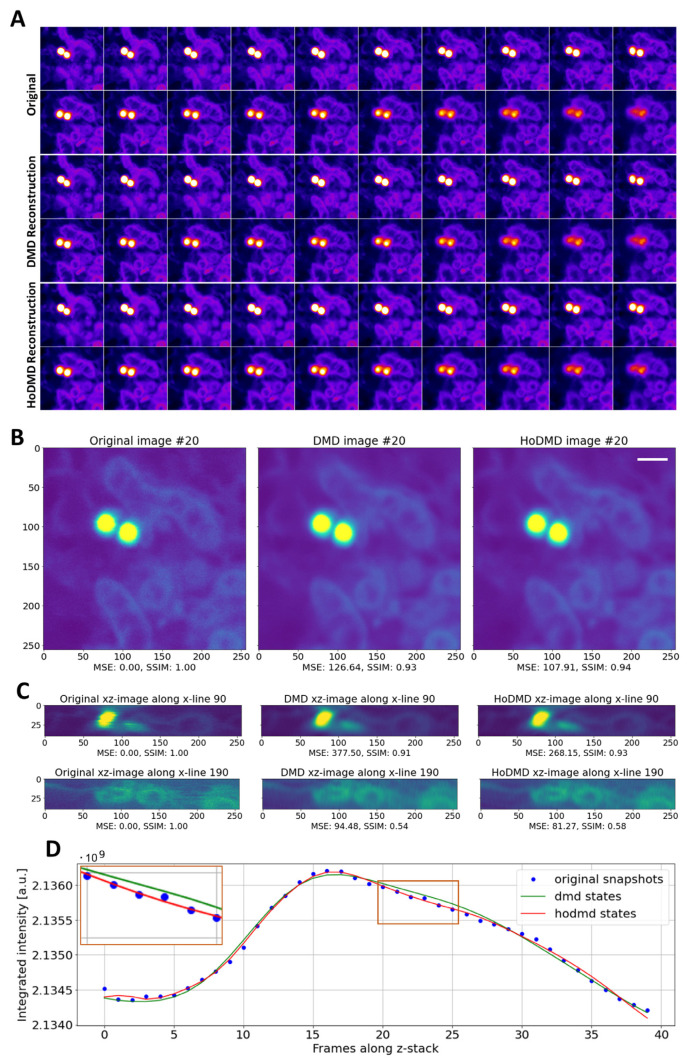
Comparison of DMD and HoDMD in reconstructing 3D-STED image stacks. Human astrocytes were labeled with Nile Red and imaged on a STED microscope in 3D. Montages of unprocessed MP (upper row, ‘Original’), DMD reconstruction (middle row), and HoDMD reconstruction (lower row) are shown with zoomed box underneath each row in (**A**). Comparison of a selected frame (#20) in the original data, the DMD, and the HoDMD reconstruction with calculated MSE and SSIM relative to the original image (**B**). Bar, 1 µm. Two selected xz-profiles at y-position 90 and 190 are shown for the original image stack, as well as for the DMD and HoDMD reconstruction, respectively (**C**). All images are identically scaled between 0 and 250 in a 32-bit format. The integrated intensity is shown for the original snapshots (blue symbols), the DMD reconstruction (green line), and the HoDMD reconstruction with a delay of *d =* 4 (red line) (**D**). The inset is a zoomed-in version of the rectangular box to highlight differences between DMD and HoDMD reconstructions.

**Figure 7 sensors-24-02096-f007:**
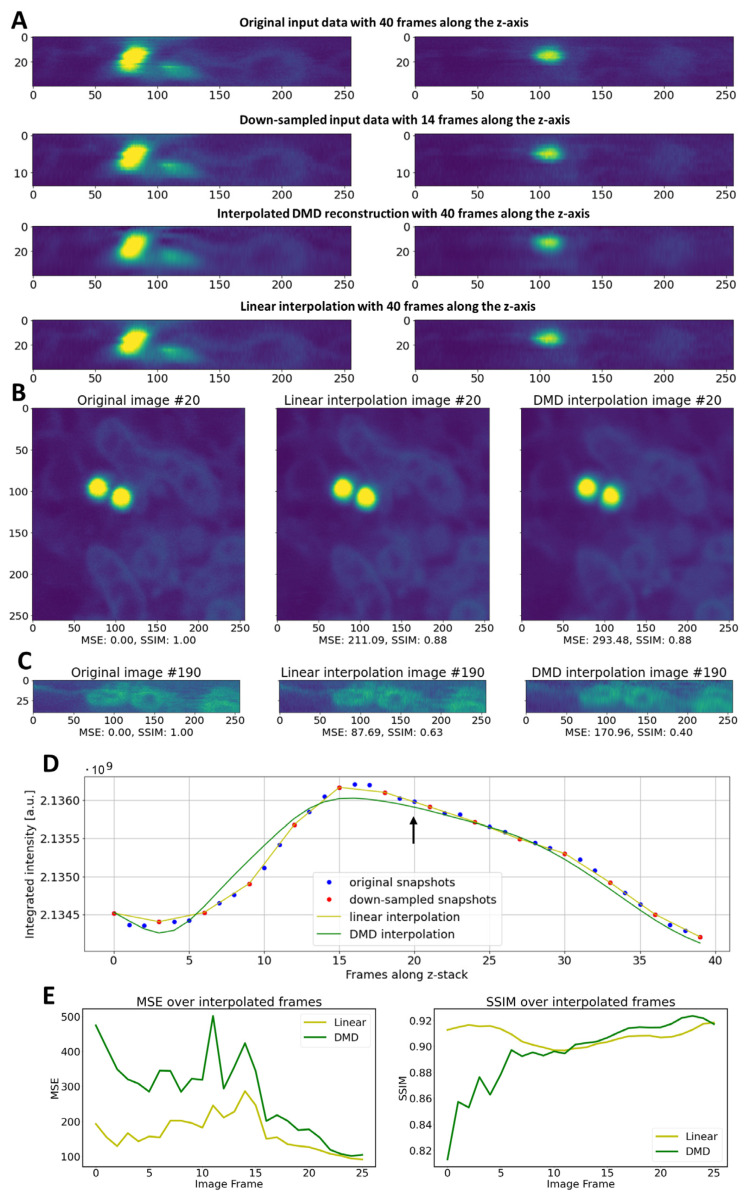
DMD is able to predict image frames, allowing for image interpolation in 3D-STED microscopy. The original image stack consisting of 40 frames was down-sampled to 14 frames by removing every third image (**A**). DMD is able to recreate the 40 frames in the reconstruction by predicting the missing frames, resulting in an only slightly lowered image quality compared to the reconstruction of the full image stack (**B**). This is also visible along the optical axis (**C**). The integrated intensity along the optical axis of the full 3D-STED image stack (blue symbols) and the down-sampled image stack (red symbols) is compared to the intensity of the linearly interpolated down-sampled stack (yellow line) and of the DMD reconstruction/interpolation of this stack (**D**). The arrow points to frame 20 shown in (**B**), which is predicted by DMD, as it was not part of the input data. Bar, 1 µm. MSE and SSIM for the interpolated frames are shown along the optical axis (**E**).

## Data Availability

All data analyzed during this study are included in this published article (and its Appendix A files). The original datasets generated and/or analyzed within the current study are available in the GITHUB repository, DanielW-alt/MP-DMD (github.com).

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
