# Peer review of "Dynamic Mode Decomposition of Multiphoton and Stimulated Emission Depletion Microscopy Data for Analysis of Fluorescent Probes in Cellular Membranes"

_sensors, 2024, doi:10.3390/s24072096_

Round 1

Reviewer 1 Report

Comments and Suggestions for Authors

In the manuscript "Dynamic mode decomposition of multiphoton and stimulated emission depletion microscopy data for analysis of fluorescent probes in cellular membranes", the adoption of DMD and HoDMD efficiently removed image noise and improved image recovery of fluorescence microscopy data, live cell imaging and other microscopy modalities, providing a post-processing method for microscopy data and a high-fidelity image reconstruction. Compared with the previously published materials, the authors adopted a more persuasive statistical analysis to evaulate the imaging results. The conclusions were consistent with the evidence and the references appropriate.

There is a question on figure citation should be paid attention. As "all MP imaging data of fluorescent sterols, DHE and TopFluor-cholesterol has been generated and described in the previous publications, and the image data analyzed is entirely from these studies", the relevant figures, such as Figure 2A, should be labelled citation in the figure legend.

Author Response

Response: We thank the reviewer for the positive assessment of our work. References to the original studies have been added to the relevant figure legends in the revised manuscript.

Reviewer 2 Report

Comments and Suggestions for Authors

The authors employed dynamic mode decomposition (DMD) and higher-order DMD (HoDMD) to reconstruct and denoise multiphoton and stimulated emission depletion microscopy data of lipid probes. This approach holds promise for reducing the number of image acquisitions, thus minimizing light exposure on biological samples while preserving image quality. The manuscript is well-presented and fits within the scope of the Sensors journal. Minor comments for improvement are provided below:

1.     In Figure 2b-c, please consider increasing the text size for better readability.

2.     Kindly include a Conclusion section to provide a summary of the findings.

3.     It is suggested to separate Appendix A-F as a Supplementary Materials file for clarity

Comments on the Quality of English Language

Minor editing of English language required

Author Response

  1. In Figure 2b-c, please consider increasing the text size for better readability.

Response: We thank the reviewer for the constructive comments to improve our manuscript. Figure 2B and C has been resized, such that the figure text and labels are easier to read.

  1. Kindly include a Conclusion section to provide a summary of the findings.

Response: A Conclusion section has been added in the revised manuscript.

  1. It is suggested to separate Appendix A-F as a Supplementary Materials file for clarity

Response: The Appendices have been removed from the main text and are now provided as separate document covering Supplementary Materials.

Comments on the Quality of English Language

Minor editing of English language required

Response: The revised manuscript has been proofread and remaining typos or grammar mistakes have been removed. 

Reviewer 3 Report

Comments and Suggestions for Authors

In this study, Authors demonstrated the possibilities of DMD for high-fidelity reconstruction and  denoising of MP and STED microscopy data of lipid probes in living cells. It has been shown that DMD is on par with other efficient image denoising methods. Using HoDMD, Authors were able to reconstruct and denoise also 3D MP stacks of DHE and account for small displacement of cells during image acquisitions. Moreover,   DMD of STED images of Nile Red not only provides much improved SNR  but also allows for predicting unseen image frames along the optical axis. So, this  study demonstrates the large potential of DMD for efficient postprocessing and analysis of live-cell fluorescence imaging data.

In my opinion, the article can be accepted in its presented form.

Sincerely, Reviewer

Author Response

Response: We thank the reviewer for the comment and positive assessment of our manuscript. 

Reviewer 4 Report

Comments and Suggestions for Authors

In this study, the power of DMD for high-fidelity reconstruction and denoising of MP and STED microscopy data of lipid probes in living cells. The HoDMD also enables one to reconstruct and analyze two-photon polarimetry data of TopFluor-cholesterol, thereby enabling reliable assessment of probe orientation in membranes. Finally, the DMD of STED images of Nile Red not only provides much improved SNR but also allows for predicting unseen image frames along the optical axis. This enables one to reduce the light exposure of the samples and thereby to minimize sample damage. Together, this study demonstrates the large potential of DMD and its variants for efficient postprocessing and analysis of live-cell fluorescence imaging data. While the paper is generally well-written, there are specific areas that warrant careful consideration for improvement.

1.      Line 496: The phrase " and to extract the dominant modes" should be revised to " and extract the dominant modes"

2.     Line 501: The phrase “showing that both, DMD and HoDMD should be revised to “show that both DMD and HoDMD”

3.     Line 510: The phrase “This representation should therefore also allow for predicting image frames not available in the original data.” should be revised to “Therefore, this representation should also allow for predicting image frames not available in the original data.”

4.     Line 583: The phrase “but also to interpolate missing frames.” should be revised to “but also interpolate missing frames.”

5.     Line 631- 635: There is no specified way to use ImageJ to simulate the PSF of the model.

6.     The structure of the cell membrane in page 7, the first figure is a bit simple, and the properties of the cell membrane are not fully reflected.

7.     In page 9panel 2 The parameters used for the reconstruction and denoising of the DHE-labeled cell image are not highlighted.

8.     In page 16: As shown in Figure 6, the line graph of performance comparison between the two DMD methods is not significantly different. It's a good idea to be able to provide a partial zoomed-in view of the line chart.

9.     In the A3-G diagram on page 22, there is some overlap in the sampling points of the curve, and local magnification can be considered.

In conclusion, this paper is organized well and have intact content. I recommend the publication in Sensors after a minor revision in which some necessary detail are added and novelty points are explained.

Comments on the Quality of English Language

Please check carefully for the writing.

Author Response

Response: We thank the reviewer for the appreciation of our work and for pointing out some grammar mistakes after carefully reading the manuscript.

  1. Line 496: The phrase " and to extract the dominant modes" should be revised to " and extract the dominant modes"

Response: This has been corrected as ‘…and for extracting the dominant modes’.

  1. Line 501: The phrase “showing that both, DMD and HoDMD” should be revised to “show that both DMD and HoDMD”

Response: This phrase has been changed to ‘…which shows that both, DMD and HoDMD’.

  1. Line 510: The phrase “This representation should therefore also allow for predicting image frames not available in the original data.” should be revised to “Therefore, this representation should also allow for predicting image frames not available in the original data.”

Response: This has been corrected, as suggested.

  1. Line 583: The phrase “but also to interpolate missing frames.” should be revised to “but also interpolate missing frames.”

Response: This phrase is now ‘… but also for interpolating the missing frames’.

  1. Line 631- 635: There is no specified way to use ImageJ to simulate the PSF of the model.

Response: The term for the used ImageJ plugin has been set into apostrophes and the link for download is provided in the revised manuscript. This part is now in the supplemental materials.

  1. The structure of the cell membrane in page 7, the first figure is a bit simple, and the properties of the cell membrane are not fully reflected.

Response: This is a very good suggestion. Panel A of Figure 1 has been replaced in the revised manuscript, now better reflecting the complexity of the plasma membrane (PM). In particular, a snapshot from a molecular simulation carried out by the lead author in the past hast been used in an artistic way together with a cartoon of the underlying cytoskeleton. This reflects better the architecture of the PM. In conjunction with this improvement of the graph, ‘interaction with the cytoskeleton’ has been added as bullet point in the figure and additional references have been cited in the Introduction section.

  1. In page 9panel 2 The parameters used for the reconstruction and denoising of the DHE-labeled cell image are not highlighted.

Response: Assuming that this comment refers to the PURE Denoise method, this information is now provided in the Materials and methods section at the end of paragraph 2.2, where the multiphoton imaging is described.

  1. In page 16: As shown in Figure 6, the line graph of performance comparison between the two DMD methods is not significantly different. It's a good idea to be able to provide a partial zoomed-in view of the line chart.

Response: An inset with a zoomed version of part of the graph has been added to Figure 6, panel C, as suggested by the reviewer.

  1. In the A3-G diagram on page 22, there is some overlap in the sampling points of the curve, and local magnification can be considered.

Response: An inset with a zoomed version of part of the graph has been added to this Figure, which is now Fig. S3 in the Supplementary materials.

In conclusion, this paper is organized well and have intact content. I recommend the publication in Sensors after a minor revision in which some necessary detail are added and novelty points are explained.

Response: We thank the reviewer again and want to remark that a Conclusions section has been added to emphasize the novelty (see also response to reviewer 3).